# Experimental Study on Dynamic Response Characteristics of RPC and RC Micro Piles in SAJBs

**Junfeng Cheng [1]**, **Xiaoyong Luo [1,***], **Yizhou Zhuang [2]**, **Liang Xu [3]** and **Xiaoye Luo [4]**

1   School of Civil Engineering, Central South University, Changsha 410075, China
2   College of Civil Engineering and Architecture, Zhejiang University of Technology, Hangzhou 310014, China
3   School of Civil Engineering, Chongqing University, Chongqing 400045, China
4   College of Civil Engineering, Fuzhou University, Fuzhou 350108, China
*   Correspondence: csu-luoxy@csu.edu.cn; Tel.: +86-731-8265-4329

**Abstract:** The pile foundations below approach slab in a semi-integral abutment jointless bridge (SAJB) that requires high flexibility to accommodate the horizontal cyclic deformation of approach slab generated by the girder's thermal expansion and contraction as well as earthquake action. In this paper, reactive powder concrete (RPC) and reinforce concrete (RC) micro piles were designed and fabricated. The shaking table tests on dynamic response of micro piles-soil interaction were conducted to investigate the dynamic response characteristics such as the strain time history of pile-soil system, the bending moment, and the deformation of piles. The maximum strain response of piles was observed at the buried depth of 4.2 D (D is the diameter of pile). Meanwhile, the maximum bending moments of RPC and RC piles appear at the depth of 0.64 D and 0.42 D, respectively, under the dynamic load excitation, and the peak horizontal deformation of piles were observed at pile head. It is found that the bending moment and the strain response of the RPC pile are larger than that of the RC micro pile, and increased by 40% and 98%, respectively. The RPC micro pile has better crack resistance, higher ductility, and flexural rigidity than that of the RC pile, and it can be widely used as pile foundations in SAJBs for the earthquake area.

**Keywords:** semi-integral abutment jointless bridge (SAJB); dynamic response characteristics; RPC micro pile; RC micro pile; shaking table tests; pile-soil interaction

## 1. Introduction

Jointless bridge is the one with continuous superstructure and without moveable joint (MJ) or movable deck joint (MDJ) between the superstructure and abutment. It has a link slab between the outer ends and girders. It includes the integral abutment jointless bridge (IAJB), the semi-integral abutment jointless bridge (SAJB), and the deck-extension abutment jointless bridge (DAJB) [1,2]. IAJB and SAJB are jointless bridges with integral or semi-integral abutment, and they have been widely used in the developed countries for its integrity and economy, such as North America, Europe, and Australia [3–5]. At present, although semi-integral abutment jointless bridges (SAJB) are widely used abroad for these properties composed of good integrity, low-cost, and a good comfortable driving environment to vehicles. However, SAJB has relatively poor seismic performance compared with IAJB [6,7]. Meanwhile, the longitudinal structures of bridge and the constraint structures of transverse anti-falling beams in SAJB are easier to damage under seismic load [8–13]. Therefore, Zhuang [14] and Chen [15] proposed a new type of micro piles made by concrete to improve the seismic behavior of SAJB. They were constructed in the backfill filling soil behind abutment below approach slabs, as shown in Figure 1. The micro-piles can be taken as a vertical brace structure for the approach slab to reduce the settlement and subsidence of the approach slab. In addition, the micro-piles can work as a longitudinal

elastic constraint to enhance the resistance of girder and approach slab under temperature loads and other effects. As a result, the expansion and contraction between the approach slab and road can be reduced. During the earthquake action, the micro-pile and soil interaction can dissipate the seismic energy to play a role as damping equipment and also increase the horizontal constraint to prevent the girder from failing. However, the design of the approach slab in SAJB is relatively complicated due to the elimination of the expansion joint. The pile foundations below the approach slab require high flexibility performance to accommodate the cyclic horizontal deformation of the approach slab generated by the girder's thermal expansion and contraction as well as earthquake action.

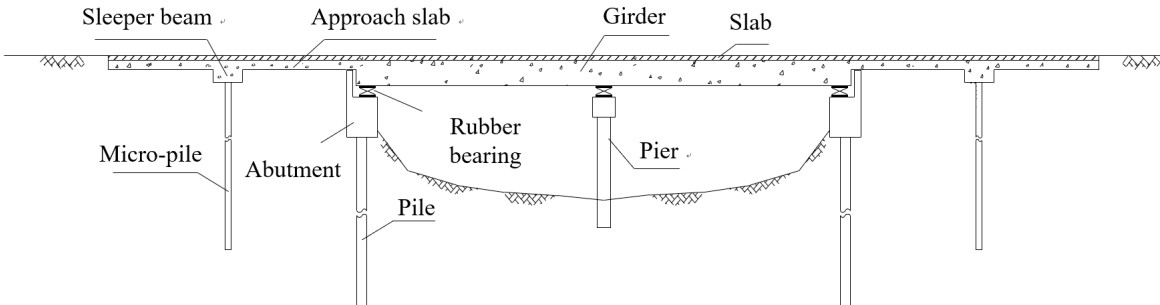

**Figure 1.** A new-type semi-integral abutment jointless bridge (SIAJB).

Micro-piles usually refer to those small-diameter piles within a diameter of 100 to 300 mm and a slenderness ratio of more than 30 [16]. Koreck et al. [17] proposed the application of micro-pile in detail, which lays the foundation for the research and development of the micro-pile in the future. At present, the studies on micro-pile are more extensive, but it mainly focuses on the testing research studies under static load or finite element (FEM) analysis, and lack the related experimental investigation under dynamic load. The most completed book on micro-pile have been published by the United States Federal Transportation Administration (DOT), and it aims to provide engineers with comprehensive technical guidance and application design of the micro-pile in practical engineering [18].

Chen [19] and Farina [20] have done some work on the dynamic responses of structure-soil interaction based on the shaking table tests. It is found that the seismic responses of structure and soil were more sensitive to input motions with richer low-frequency components. Meanwhile, the frequency spectrum characteristics of input ground motions clearly affected the lateral displacement of the structure. Furthermore, the influences of mismatches on the simulated dynamic response of a simple structural model were further investigated. Sung et al. [21] conducted the mechanical properties of single micro pile in sand under axial cyclic loading, focusing on the characteristics of side friction between the micro pile and the cast-in-situ pile. However, it is well known that accidental loads (strong earthquakes) will generate greater damage to the pile foundation of the bridge. Therefore, it is of great significance to study the dynamic response characteristics of micro pile-soil interaction under a seismic wave load. Chen [22] conducted a shaking table test on a single circular steel tube micro pile-soil system to investigate the energy consumption of the micro pile under a different counterweight load at the top of the pile, and then the energy dissipation mechanism of the micro pile under sandy soil conditions were discussed in detail. Wang et al. [23] investigated the effect of reaming on the mechanical properties of micro piles based on the shaking table test, and the energy dissipation mechanism of micro piles were obtained. However, the energy dissipation effect of ordinary concrete micro piles is not desirable. Fan [24] also studied the influences of reaming conditions on the mechanical properties of the micro pile foundation based on a quasi-static test. It was found that the 'm' method and the 'API' method were modified according to the test results, and the static *p-y* curves of micro pile-soil suitable for sandy soil conditions was deduced and proposed. In addition, there are numerous research studies based on finite element model (FEM) analysis to simulate the interaction effect between the micro pile and the soil. Based on the Winkler foundation beam model, the soil resilience distributed continuously is discretized to be the force performed by a series of discontinued beam elements with springs. This

measure can simplify the analysis of the pile-soil interaction [25,26]. The horizontal deformation and rotation can be considered in the note where the masses of pile were discretized and concentrated, while the soil surrounding the pile is simulated by a series of discretized spring-damper elements in which the stiffness of spring is determined by *p-y* curves such as proposed by National Cooperative Highway Research Program (NCHRP) [27], American Petroleum Institute. (API-RP2A) [28], and Reese [29]. Regarding all of the above, including a number of studies on micro piles-soil interaction, they mainly focus on static loads and numerical simulation investigations. However, limited research studies have been focused on the dynamic response of micro pile-interaction, and these research studies are relatively monotonous, failing to reflect the micro pile-soil system under dynamic loads and relatively poor seismic capacity under an earthquake load. To be noted, most of the piles used in above studies are ordinary concrete micro piles whose seismic performance is relatively poor. Therefore, it needs further investigation to improve the seismic performance of the micro pile-soil system in semi-integral abutment jointless bridge (SAJB).

The reactive powder concrete (RPC) micro pile is a new type of micro pile, which was usually made by ultra-high concrete. Compared with the conventional reinforce concrete (C40) pile, the RPC pile has more performances composed of ultra-high compressive strength, more high durability, and more high toughness, which, as a new type, the pile in material could satisfy the requirements of the approach slab's foundation [30]. However, there are limited research studies focused on this type of pile, especially for its dynamic response properties during seismic wave load, which leaves this applicability in doubt. In past decades, researchers mainly focused on the vertical mechanical properties and bearing capacities of RPC piles, while only a small amount of them were on the dynamic response characteristics under seismic wave excitation.

Therefore, in order to make a further study on the dynamic characteristics of the micro pile-soil interaction, the shaking table tests on dynamic response properties of reactive powder concrete (RPC) and reinforce concrete (RC) micro piles were carried out. Under the dynamic load excitation, the fundamental frequencies of micro piles were obtained. The time history of strains on the pile-soil system, the bending moments, and deformations of piles were examined in this work. It lays a foundation for the future seismic research of the micro piles-soil system and the application of micro piles in SAJBs.

## 2. Experimental Program

### 2.1. Specimen Design and Fabrication

As noted previously, it is known that the micro piles are the vertical supporting piles below approach slabs in SAJB for practical engineering. The maximum size of the structure for micro pile in real-life is about 0.1–0.3 m in diameter and 2–5 m in length. In this experiment, two micro piles were designed and fabricated according the Chinese code for Building Pile Foundations in terms of the similar principle of pile diameters 'D.' They were labeled as RPC and RC, respectively. The shaking table tests on the dynamic response of full-scale micro pile-soil interaction were carried out in this work. Therefore, the pile model is the same as the practical situation, its geometrical scale is 1:1, and the stress level ratio is 1 [31]. The RPC and RC specimens are 2.20 m in long, and 100 mm in diameter. The RPC micro pile model is a new type of pile made by new concrete materials, which is composed of steel fiber, fly ash, and other materials by mixing. Furthermore, steel fiber of 2% concrete content (mass fraction) in diameter of 8 mm was mixed into the mix design of the RPC micro pile, which enhanced significantly the strength, stiffness, and toughness of the RCP pile without coarse aggregate. In this case, the water binder ratio of 0.18, the silica fume with 0.3 of concrete, and fine sand with 1.2 times (mass fraction) of concrete were employed as the mixtures. In addition, the mechanical parameters of its test cube as compressive strength of 141.6 MPa, Poisson ratio of 0.25, and an elastic modulus of $4.41 \times 10^4$ MPa were measured after 28 days. Meanwhile, the designed strength grade of the RC micro pile was C40. It serves as a comparative specimen of the RPC micro pile. Therefore, the material properties and other

parameters of RPC and RC model piles are also summarized in Table 1. The transverse reinforcement of two micro piles was 6 mm in diameter and its spacing was 200 mm, and 6Φ10 mm in diameter rebars were selected as the longitudinal reinforcement bars. The reinforcement ratio of each micro pile was 6%, which meets the minimum reinforcement ratio of 0.65% of the Chinese code. The process of fabrication and reinforcing bars of micro piles are shown in Figure 2a,b, respectively. The PVC pipe with 100 mm in diameter was used as the mold of micro piles before the construction of model piles.

**Table 1.** Parameters of micro piles.

| Specimen | Diameter (mm) | Moment of Inertia $I$ ($\times 10^{-5}$ m$^4$) | Modulus of Elasticity $E$ ($\times 10^4$ MPa) | Relative Pile Length $z$ (m) | Flexural Rigidity (kN·m$^2$) | Section |
|---|---|---|---|---|---|---|
| RPC pile | 100 | 0.49 | 4.41 | 3.49 | 216.7 | circular |
| RC(C40) pile | | | 3.25 | 3.78 | 159.3 | |

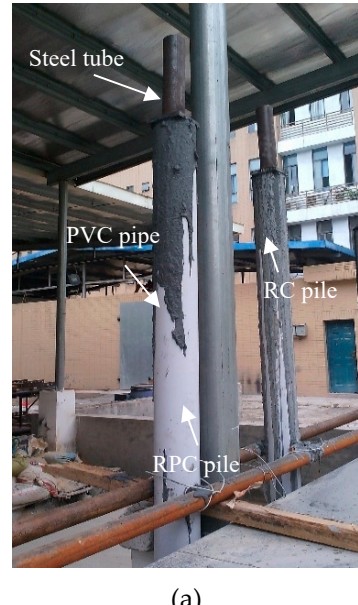

(a)

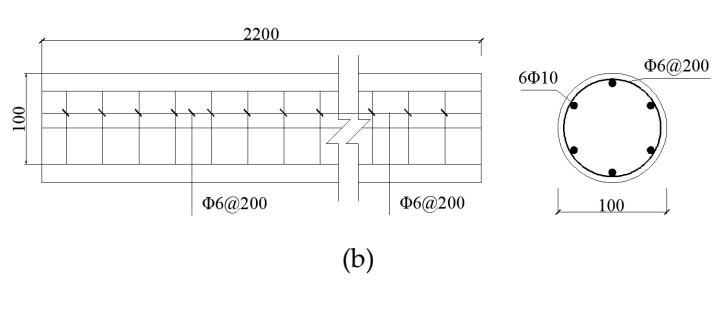

(b)

**Figure 2.** Micro pile models (unit: mm). (**a**) Fabrication of micro-piles. (**b**) Reinforcement drawing.

The inertia mass block was embedded at the pile head to replace the approach slab in SAJB supported by micro-piles [32]. Assuming the actual dimensions with the length of 3 m and the thickness of 0.2 m for each unit width of the approach slab, and this approach slab was supported by three micro piles in a row at the end of the abutment. Meanwhile, the supporting influence of soil lower the slab on the approach slab, which was neglected, and the weight of the concrete was calculated as 2.2 ton/m$^3$. As a result, the inertia mass at the top of each micro-pile was 220 kg, and the mass block was made by welding the counterweight iron brick, as shown in Figure 3a. In addition, it is convenient that the fixed connection of mass block and pile head, as well as a hole with a diameter of 60 mm was reserved in the center of the mass block, and the steel tube was welded on the pile head along the axis direction of the pile. Therefore, the mass block was embedded on the outer wall of the steel tube along the length direction of the steel tube, and the mass block was fixed by welding to the steel tube.

### 2.2. Soil Container and Soil Parameters

The sand used in this test was from Min River in the Fujian province, China, where a number of SAJBs were constructed. The natural sandy soil was dried to reduce water content and filtered to remove impurities as well as large particles before the test. The nonuniform coefficient Cu = 3.16, which

means the soil was basically homogeneous [33]. Poisson's ratio was calculated by the Kulhawy method (1990) [34]. Its average SPT blow counts of the sand were 12. According to ASTM Standards [35], it is classified as medium sand. The parameters of sand are listed in Table 2.

**Table 2.** Physical and mechanical parameters of sand.

| Water Content $\omega$ (%) | Density $\rho$ (g/cm$^3$) | Void Ratio $e$ | Cohesive Strength $c$ (kPa) | Internal Friction Angle $\varphi$ (°) | $C_u$ | Poisson Ratio $v$ |
|---|---|---|---|---|---|---|
| 2.2 | 1.98 | 0.80 | 4 | 32 | 3.15 | 0.3 |

The boundary condition is a crucial factor to investigate the dynamic response of micro pile- soil interaction in this study. Theoretically, the larger the soil box, the more accurate the results. However, as a consideration of the lab space and economy, the dimensions of the soil box are usually of a limited size. Lou et al. [36] conducted shaking table experiments and numerical analysis on the structure-soil interaction. It was concluded that, when the plan dimension of the soil box is five times that of the structure, the experimental errors caused by the boundary condition are small. In addition, the FEM method [37] was employed to analyze the soil box and determine the dimensions of the soil box. It indicated that the influence of the boundary can be neglected when the radius of the soil box is larger than five times of the pile's diameter.

Based on the above results, a rigid square steel box with a width of 2.0 m, length of 2.0 m, and height of 2.1 m and wall thickness of 10 mm was employed to constrain the soil, as shown in Figure 3a. The soil box was welded by four steel plates of 10 mm thickness, and each steel plate was stiffened by a number of diagonal braces around its outside edge. In order to facilitate sand filling and unloading, three square steel plate doors with the dimensions of $200 \times 200 \times 10$ mm were designed on three sides, respectively, at the lower outside surface of the steel plate, and these doors were closed when sand is filling into the soil box. The length and width of the soil box are approximately 20 times diameter of the micro-pile, which is sufficient to consider the boundary condition. The bottom of this box was welded to a bottom plate with dimensions of $2200 \times 2200 \times 10$ mm. The bottom plate was then anchored on the shaking table surface by high-strength bolts through numerous screw rods with 16 mm in diameter to achieve a fix base condition, as shown in Figure 3a. To eliminate the deformation of the steel box for too small of a stiffness during seismic wave loading, a total of eight inclined angle steels were used as support on the outer side of the steel box. Four 150–mm long, steel angles were welded inside the box to the center of the bottom plate to fix the micro pile at its end tips before the steel box is filled with sand, which is shown in Figure 3b.

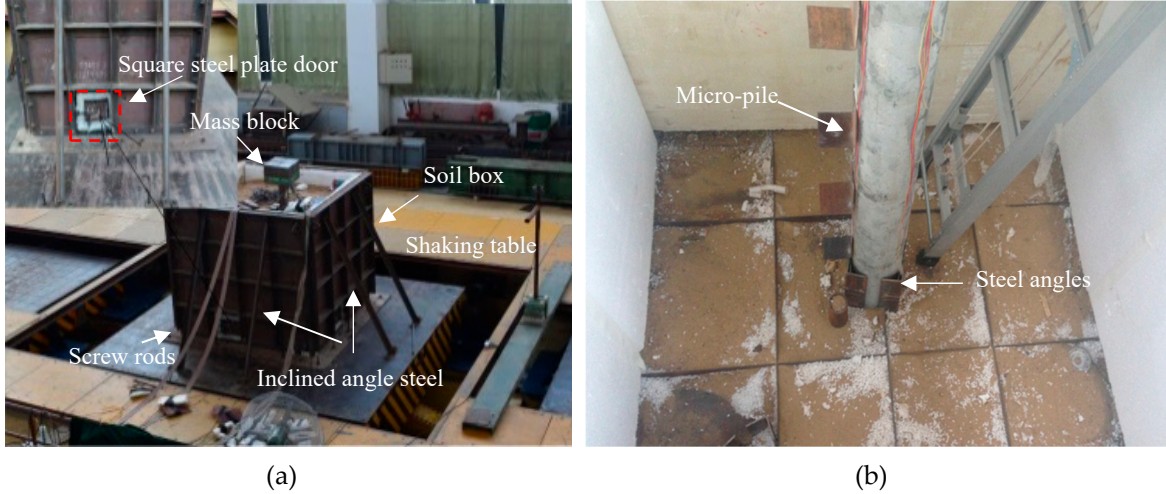

(a)　　　　　　　　　　　　　　　　(b)

**Figure 3.** Details of the soil box and layout of pile. (**a**) Steel box. (**b**) Model pile in box.

Two different boundaries were employed in the inside of the sand box, and some angle steels were welded into the bottom of the box to form an anti-skid compartment. The relative movement between the bottom sand and the box can be prevented. In addition, the flexible boundary was established inside of the sand box by sticking 10–cm thickness polyethylene foam board to the inner of the steel box walls. Therefore, the boundary condition of the micro pile-soil interaction can be as closed as possible to the real situation [38–40]. In this experiment, the embedded depth of each micro-pile was 1.8 m, which left the above-ground part of 0.4 m, as reference literature [41]. After the micro pile was in position, it began to fill the box with soil up to 1.8 m of height, as shown in Figure 4a. The soil filled in the box was tamped by each layer of 20 cm height until the embedded depth is 1.8 m.

Elastic pile is widely used in SAJB to satisfy the longitudinal deformation of the superstructure under the temperature load or earthquake load. According to a technical code for the building pile foundation (China) [42] and literature [43], it is found that, when the equivalent length ($\bar{l}$) is larger than 2.5, the pile belongs to the elastic pile. The equivalent length is calculated by the following.

$$\bar{l} = \alpha l_0 \tag{1}$$

$$\alpha = \sqrt[5]{\frac{mb_0}{EI}} \tag{2}$$

$$b_0 = 1.5b + 0.5 \tag{3}$$

where $l$ is the equivalent length of the micro pile (m), $\alpha$ is the relative flexibility coefficient, $l_0$ is the embedded depth of pile (m), $m$ is the coefficient of horizontal subgrade reaction (MN/m$^4$), $b_0$ is the calculating width of pile (m), $E$ is the elastic modulus of the pile (kN/m$^2$), $I$ is the moment of inertia of pile (m$^4$), and $b$ is the diameter of pile (m).

Based on Equations (1)–(3), the equivalent length ($\bar{l}$) of the RPC and RC micro-piles are 3.49 m and 3.78 m, respectively, which meets the requirement of the elastic pile in the code.

*2.3. Measuring Points Layout*

In this experiment, sensors were placed at specific locations to measure the strain, the displacement, and the acceleration of micro piles. A total of 20 strain gages were attached to the outmost edges (front and back sides) of each pile symmetrically in the loading direction to measure the strain distribution, respectively. These gages started from the bottom of the soil to above the soil surface of 212.5 mm, symmetrically, with nine equal spacings of 212.5 mm and a spacing of 100 mm. They were labeled as S1 to S20, as shown in Figure 4a. Meanwhile, eight acceleration sensors were employed to measure the acceleration at the outmost edges of each pile in the loading direction, respectively. In this case, six accelerometers started from the bottom of the soil to soil surface with six equal spacings of 300 mm. They were labeled as A1 to A6. There are two accelerometers utilized to measure the acceleration at the upper surface of the mass block and shaking table, respectively (marked A7 and A8), as plotted in Figure 4b. In addition, two displacement meters were arranged on the lower surface of the mass block and on the upper surface of the shaking table to measure the displacement of the pile head and shaking table, respectively, and labeled as D1 and D2, as shown in Figure 4c.

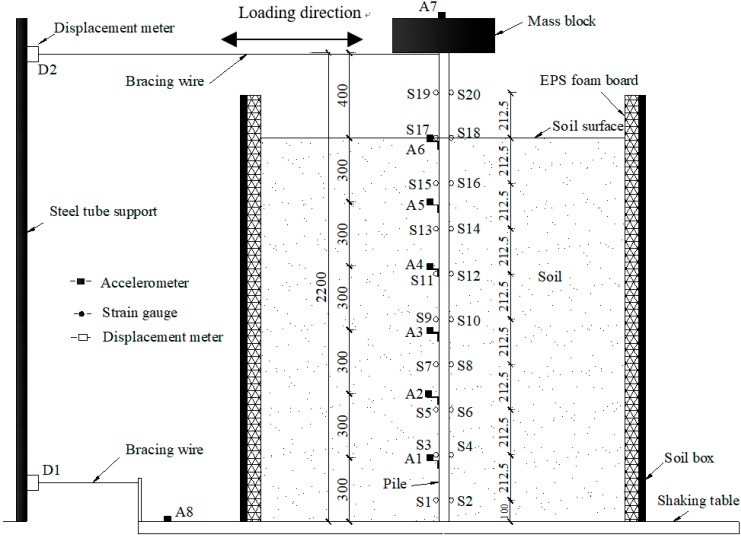

(a)

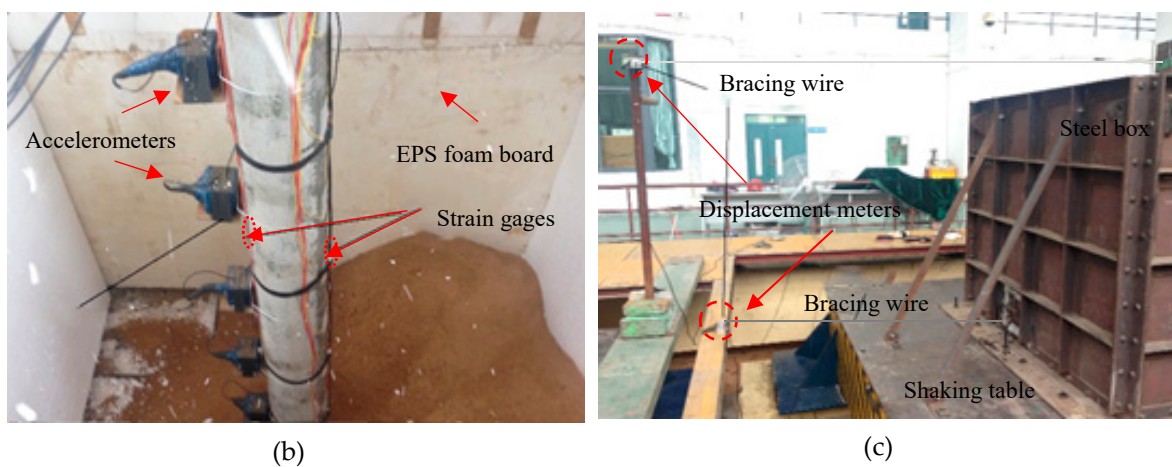

(b)　　　　　　　　　　　　　　　　　　　　(c)

**Figure 4.** Layout of sensors for each micro pile. (**a**) Layout diagram of sensors (unit: cm). (**b**) Accelerometers and strain gauges. (**c**) Displacement meters.

### 2.4. Loading Device and Scheme

To be noted, previous studies [36,44,45] proposed that the sine wave was the most suitable wave to obtain the dynamic response characteristics of the pile-soil system. Therefore, the sinusoidal wave is the main load selected in this test to investigate the dynamic response characteristics of micro pile-soil system. In addition, the dynamic response characteristics of the micro pile-soil interaction system under the EI-Centro wave, the Kobe wave, and artificial wave loads were also discussed, respectively. The acceleration time history of the El-Centro wave, the Kobe wave, and the artificial wave load are plotted in Figure 5. According to the provisions of seismic fortification intensity, the design basic seismic acceleration is in the code for seismic design of GB50011-2010 (China) [46]. It is found that the seismic fortification intensity of the project was 7 degrees, and the design basic acceleration was 0.15 g. Therefore, the experimental study on dynamic response characteristics of RPC and RC micro pile-soil interaction were conducted in the structural laboratory based on a three-array system of the earthquake simulation shaking table.

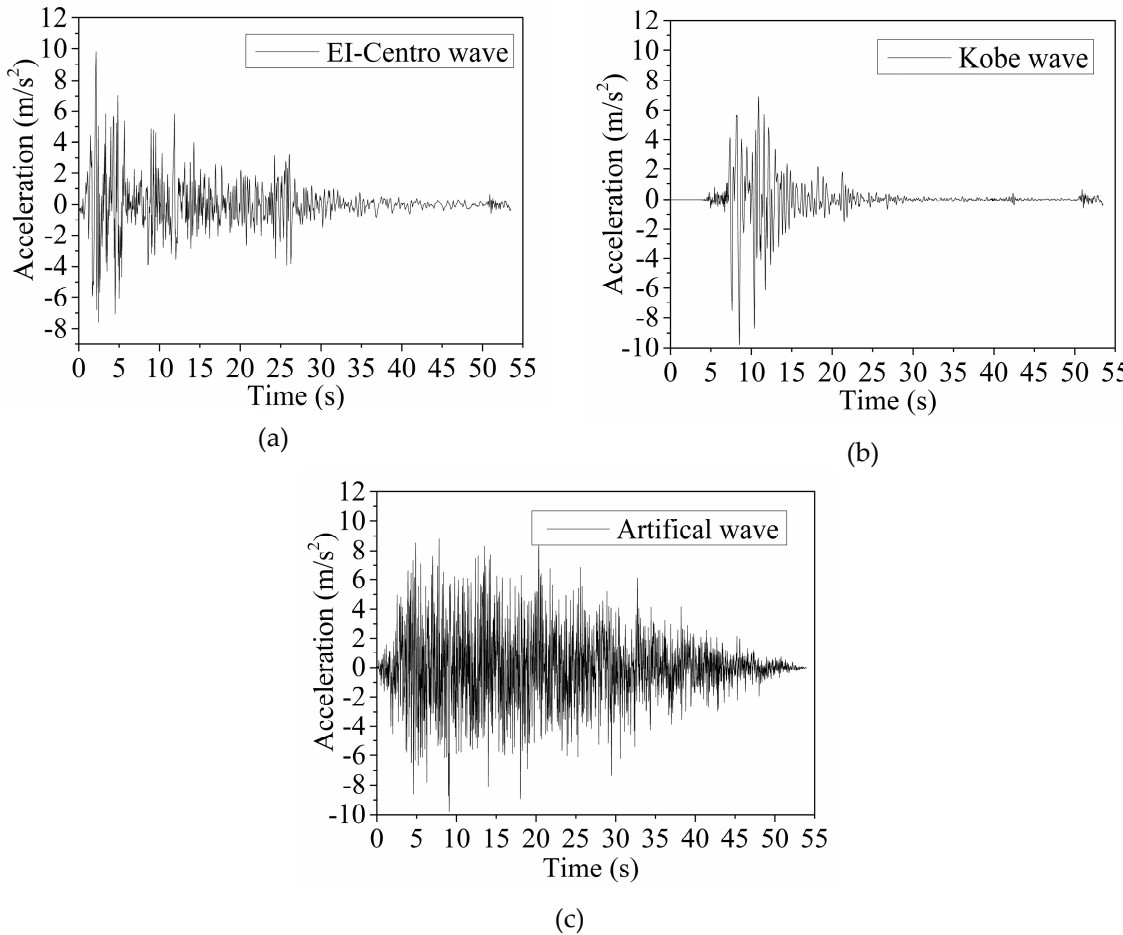

**Figure 5.** Acceleration time history of seismic wave loads. (**a**) EI-Centro wave. (**b**) Kobe wave. (**c**) Artificial wave.

As mentioned previously, a total of eight loading cases were employed as the earthquake load to examine the dynamic response characteristics of RPC and RC micro piles in this paper, as summarized in Table 3. The sine wave loads in peak acceleration of 0.15 g have different frequencies such as 1 Hz, 2 Hz, 4 Hz, 8 Hz, and 16 Hz. Before each change of the sine wave frequencies, white noise scanning was adopted to obtain natural frequencies of the pile-soil system, as shown in Table 3 (from case 1 to case 5). Meanwhile, several typical seismic waves including the EI-Centro wave, the Kobe wave, and the artificial wave loads with peak acceleration of 0.15 g, respectively, are listed in Table 3 (from case 6 to case 8).

**Table 3.** Test loading cases.

| Case | Seismic Wave | Peak Acceleration (g) | Frequency (Hz) | Case | Seismic Wave | Peak Acceleration (g) | Frequency (Hz) |
|---|---|---|---|---|---|---|---|
| 1 | White noise | | | 5 | White noise | | |
| | Sine wave | 0.15 | 1Hz | | Sine wave | 0.15 | 16 Hz |
| 2 | White noise | | | 6 | EI-Centro wave | 0.15 | |
| | Sine wave | 0.15 | 2 Hz | | | | |
| 3 | White noise | | | 7 | Kobe wave | 0.15 | |
| | Sine wave | 0.15 | 4 Hz | | | | |
| 4 | White noise | | | 8 | Artificial wave | 0.15 | |
| | Sine wave | 0.15 | 8 Hz | | | | |

## 3. Experimental Results

### 3.1. Dynamic Characteristics

Figure 6 plots the spectrum analysis results at the top of RPC micro pile under white noise waves, where Figure 6a shows the spectrum analysis results under the first white noise wave (case 1), and Figure 6b shows the fifth white noise (case 5). As seen in Figure 6, it was found that these frequencies corresponding to peak amplitude at the top of the RPC micro pile system are closed to 5 Hz and 16 Hz, respectively, and they are more clear than that of others' frequency. It was observed that the first and second natural frequency of the RPC micro-pile system were 4.74 Hz and 16.31 Hz, respectively. In addition, it was also found from Figure 6 that the natural frequency of the RPC micro pile system increased significantly after several times of repeated white noise. The reason is that the increase of sand compactness under a shaking table test, and the increase of the pile-soil interaction is significant. Meanwhile, the natural frequencies of the RC micro pile under repeated white noises were also obtained by the spectrum analysis, and the first and second natural frequencies of the RC micro-pile system were 3.62 Hz and 15.46 Hz, respectively. It was indicated that the natural frequencies of the RPC pile are larger than that of the RC micro pile.

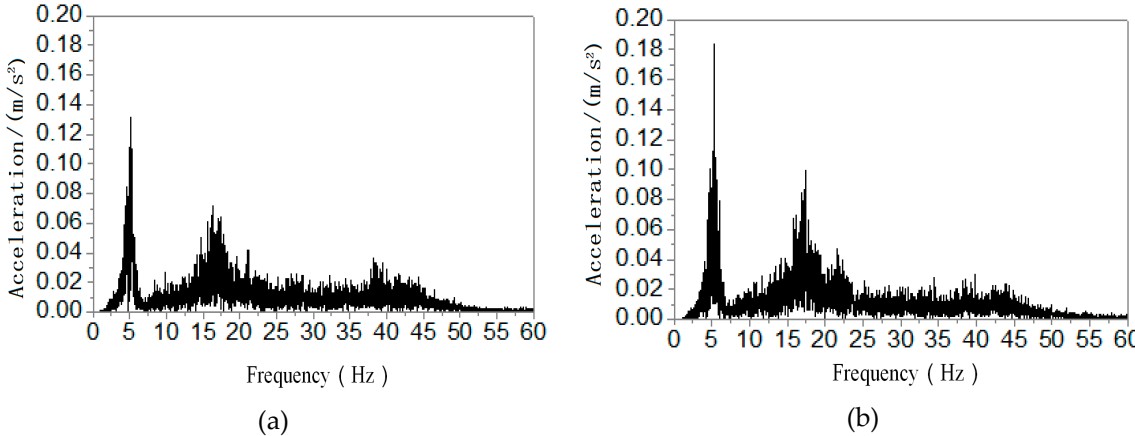

(a)         (b)

**Figure 6.** Spectrum analysis results of the reactive powder concrete (RPC) pile-soil system under white noise. (**a**) The first white noise and (**b**) the fifth white noise.

### 3.2. Strains Responses

A total of three typical strain measuring points were monitored to investigate the strain time history response of piles under the case 3 load (sine wave of 4 Hz frequency with amplitude of 0.15 g). These points were located at 212.5 mm (2.1 D) above the soil surface, at soil surface (0 D), and 425 mm (4.2 D) below the soil surface, respectively. The strain time history response of RPC and RC piles at above three points under the sine wave load are summarized in Figure 7. As seen in Figure 7, the strain time history responses of three strain measuring points at the micro-pile body are identical basically, and the strain time history response of the pile at 4.2 D below the soil surface is larger than that of the other two measuring points. It is observed that the strain response of the soil surface measuring point is second, and that of the above soil surface of 2.1 D is the minimum. It is found that RPC and RC piles are basically in the elastic stage after shaking the table test. Compared with Figure 7a,b, it is also found that the maximum strain of the RPC micro pile reaches $200 \times 10^{-6}$. However, the ultimate strain of the RC (C40) pile are between $70 \times 10^{-6}$ and $100 \times 10^{-6}$. Therefore, it indicated that the RPC micro pile has better crack resistance and deformation capacity than of the RC (C40) micro pile.

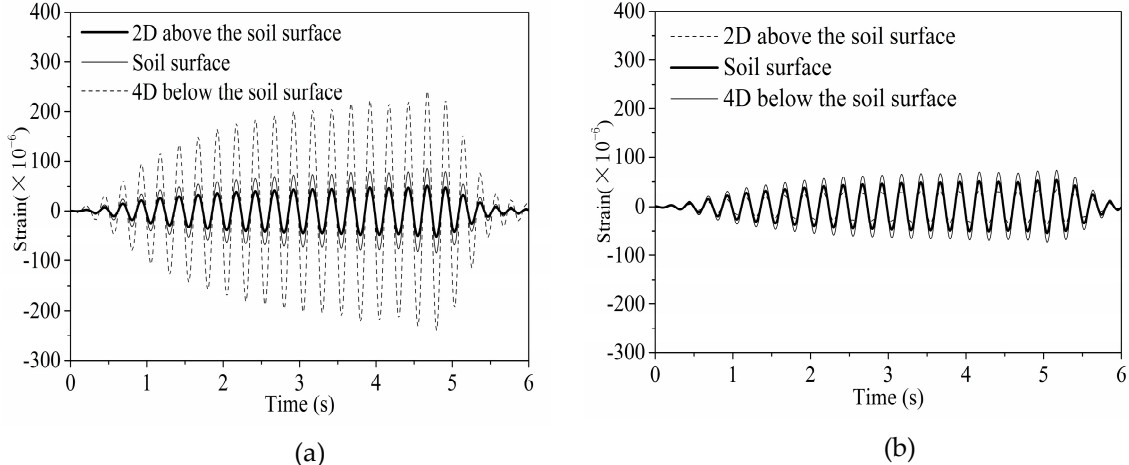

**Figure 7.** The strain time history response of micro piles. (**a**) RPC micro pile. (**b**) RC micro pile.

### 3.3. Bending Moment of Piles

The bending moment *M* of micro piles during the elastic range can be back-calculated from the measured strain under the assumption that the plane section remains plane. It is calculated by the following formula.

$$M = \frac{EI(\varepsilon_l - \varepsilon_r)}{D} \tag{4}$$

where *E* is the elastic modulus of concrete (kN/m$^2$), *I* is the moment of inertia for section (m$^4$), *D* is the diameter of pile (m), and $\varepsilon_l$ and $\varepsilon_r$ are the measured tensile and compressive strains at the outmost edges of the section.

#### 3.3.1. The Influence of Frequencies

In order to discuss the influence of different frequency loads on the bending moment of piles, the bending moments of RPC and RC micro piles under sinusoidal wave loads with different frequencies (from case 1 to case 5) are shown in Figure 8. As seen in Figure 8, the bending moment distribution laws of piles along the buried depth are basically consistent. It is noteworthy that the bending moments of piles increase linearly from the pile head to the soil surface under different frequency sinusoidal waves.

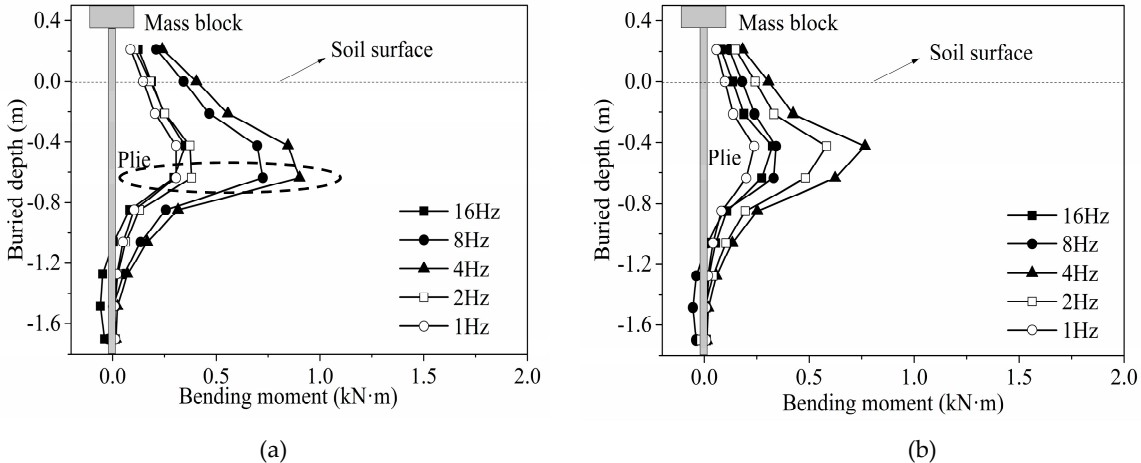

**Figure 8.** Bending moments of micro piles under different frequency sinusoidal wave loads shows (**a**) RPC micro pile, and (**b**) RC micro pile.

The bending moments of RPC and RC piles increase continuously with the rise in buried depth and reaches the maximum positive value at the buried depth of 0.637 m (6.4D) and 0.425 m (4.3D), respectively. Then, the bending moments of piles gradually diminish to 0 kN·m or even a slight negative value as the further one increases the buried depth. It is shown that the influences of elastic modulus and ductility of pile body materials on the bending moment of piles are clear, and the peak bending moment and the corresponding buried depth increases to a certain extent, as the increase of the elastic modulus.

In addition, it is also found that the bending moments of piles along the buried depth first increase and then decrease as the frequencies increase the sinusoidal waves. In this case, the bending moments of the RPC pile are 0.31 kN·m, 0.38 kN·m, and 0.92 kN·m, respectively, under the frequencies of 1 Hz, 2 Hz, and 4 Hz, and the bending moments are 0.73 kN·m and 0.30 kN·m under harmonic loads with 8 Hz and 16 Hz, as drawn in the ellipse dashed in Figure 8a. However, for the RC pile, the bending moment also reaches the maximum under the frequency of 4 Hz, and the follow is that of the 2 Hz frequency load, as plotted in Figure 8b. It can be seen that the bending moments ordering of RPC and RC micro piles are different under the same frequency loads. The reason is that the first natural frequency of the RC pile is 3.62 Hz, and the bending moment response of the pile is significant when the dynamic load is close to the frequency of 2 Hz or 4 Hz. However, the first natural frequency of RPC is 4.74 Hz, so the bending moment responses are more significant under the frequency loads of 4 Hz and 8 Hz. In addition, compared to Figure 8a,b, the maximum bending moment of RPC and RC piles are 0.92 kN·m and 0.76 kN·m, respectively. It is noted that the bending moment of the RPC pile is greater than that of the RC pile, and increased by 21%.

### 3.3.2. The Influence of Seismic Waves

The bending moments of RPC and RC piles under different seismic loads in cases 6, 7, and 8 (EI-Centro wave, Kobe wave, and artificial wave) were also output and plotted in Figure 9a,b, respectively. The bending moment distribution laws under different seismic wave loads are also basically consistent for each pile. It can be seen that the bending moments of piles above the soil surface decrease linearly from the soil surface to the top of the piles, and the bending moments of RPC and RC piles increase rapidly with the increase of buried depth and reach the maximum positive value at the buried depths of 0.637 m (6.4D) and 0.425 m (4.3D), respectively. Furthermore, the bending moments of piles gradually diminish to 0 kN·m as the buried depth increases. The reason for the different buried depths at the corresponding peak bending moment of piles also resemble what was mentioned previously. It can be seen that the elastic modulus and flexural rigidity of the RPC pile are larger than that of the RC pile, and that the peak bending moments and corresponding buried depths are larger than that of the RC pile.

As seen in Figure 9, it was also found that the peak bending moments of RPC and RC micro piles under different seismic loads in cases 6, 7, and 8 are 1.01 kN·m and 0.73 kN·m, 0.49 kN·m, and 0.40 kN·m, 1.57 kN·m, and 1.11 kN·m respectively. Therefore, it concluded that the bending moment of micro piles under an artificial wave load is significantly greater than that of the other two seismic loads. The bending moment of the pile under the Kobe wave load is smallest, and that of EI-Centro follows. Accordingly, the influence of different seismic wave loads on the bending moment of piles is remarkable, and the effect of artificial wave load is significant. Besides, the bending moment of the RC pile is smaller than that of the RPC pile, which is decreased by 40%. It was because the flexural rigidity of the RPC micro pile is larger than that of the RC pile, and the bending moment of piles increases significantly with the increase of flexural rigidity under seismic wave loads.

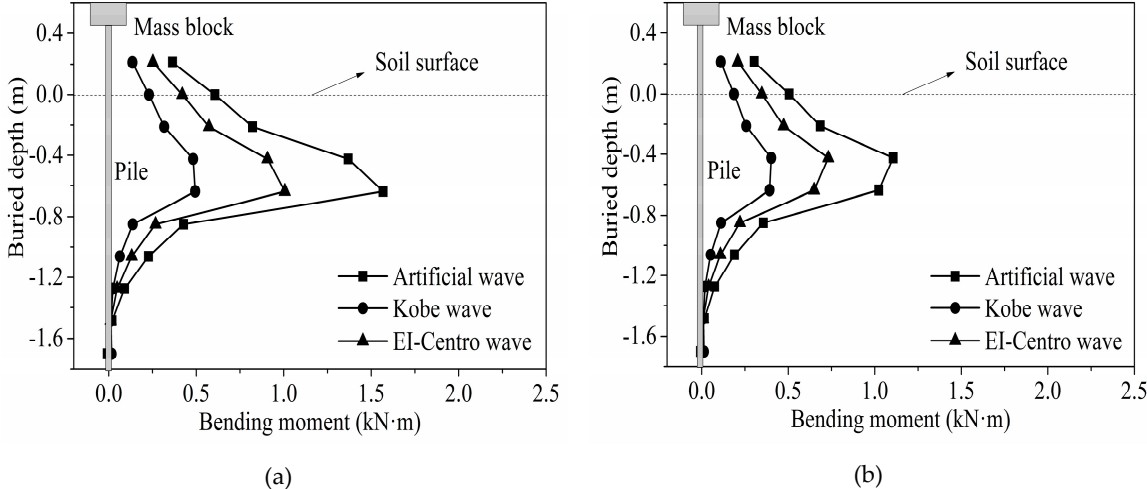

**Figure 9.** Bending moments of piles under different seismic waves loads. (**a**) RPC micro pile. (**b**) RC micro pile.

### 3.4. Deformation of Piles

As noted previously, the deformations of the pile along its length are usually back-calculated by the recorded strains. In this process, it is assumed that the pile is a flexible foundation beam. Based on the Euler Bernoulli beam theory, the deformations of the pile are derived as follows [29].

$$\varepsilon_t - \varepsilon_c = \Delta\varepsilon = \frac{M(z)}{EI} \tag{5}$$

$$M(z) = \frac{EI}{D}\Delta\varepsilon \tag{6}$$

$$y'' = -\frac{M(z)}{EI} \tag{7}$$

$$y(z_i) = \frac{EI}{D}\int_0^{z_i}\left[\int_0^{z_i}\Delta\varepsilon dz\right]dz + C_1 z_i + C_2 \tag{8}$$

where $E$, $I$, $D$, $\varepsilon_t$, and $\varepsilon_c$ are the same with Equation (4). $y$ is the deflection of the pile along its length at location $z$ (m), $z$ is the relative distance from the pile head to the point of interest (m), $C_1$ and $C_2$ are integral constants and $C_1 = EIy'(0)$, and $C_2 = EIy(0)$, and $y'(0)$ and $y(0)$ are the angle and lateral displacement of the pile head, respectively. The horizontal deformation $y(z_i)$ at different depth $z_i$ of piles can be obtained by assuming that the strain of the piles between the two adjacent measuring points changes linearly.

In order to compare the deformations of the micro pile between measured and calculated results, Figure 10 plots the deformation-time history of the pile head measured results and calculated results by the back-calculated method when the pile subjected to the sinusoidal wave of 4 Hz frequency with an amplitude of 0.15 g.

As seen in Figure 10, it is observed that the test results are larger than the results of the back-calculation method. It is known that the results of the back-calculation method calculated by the measured strains are closed to that of measured results to calculate the deformations of piles during the elastic stage. However, there is a clearer difference between the above two results when the amplitude load increases and reaches a certain value. As mentioned previously, the RPC and RC piles are basically in the elastic stage after the shaking table test. Therefore, this clearly indicates that a small amount of rigid body displacement was generated for the micro-pile in the process of larger

amplitude acceleration excitation. The displacements obtained by the test measured including parts of rigid body displacement.

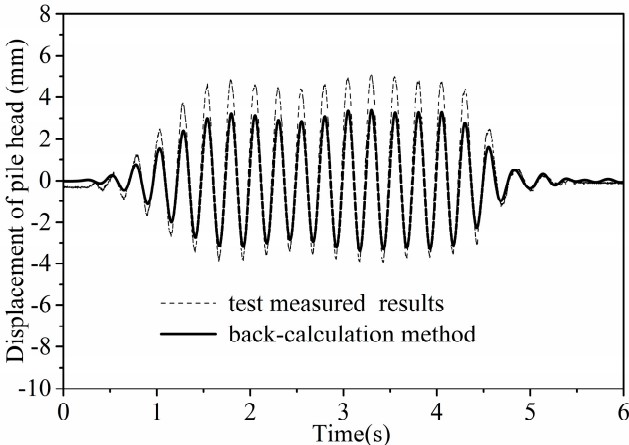

**Figure 10.** Displacement time history curves of the reactive powder concrete (RPC) pile at the pile head.

### 3.4.1. The Influence of Frequencies

Figure 11 shows the horizontal deformations of RPC and RC micro piles under sinusoidal wave loads with different frequencies (from case 1 to case 5). As seen in Figure 11, the horizontal deformation distribution laws of the pile along its buried depth are mainly consistent for each pile. The maximum deformations of piles under different frequency loads are observed at the pile head. It can be seen that the horizontal deformations of piles above the soil surface decrease linearly from the pile head to the soil surface under different sinusoidal wave loads. The horizontal deformations of the pile body decrease significantly with the increase of buried depth. Then, the bending moments of piles gradually diminish to 0 mm or even become a slightly negative value as the buried depth continuously increases.

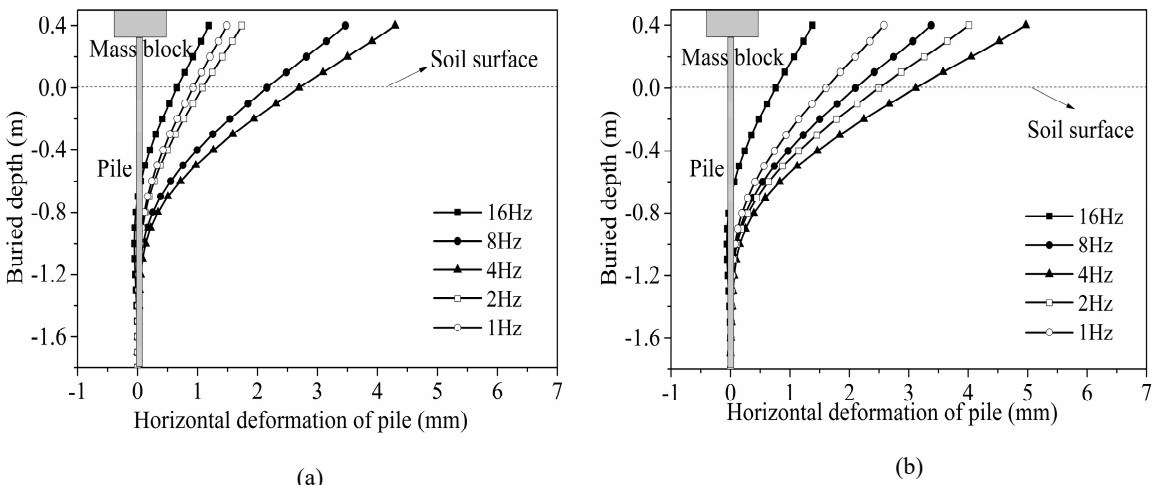

**Figure 11.** Horizontal deformations of piles under different frequencies loads. (**a**) RPC micro pile. (**b**) RC micro pile.

To be noted, it can be clearly seen from Figure 11 that the maximum horizontal deformations of piles under different frequency sine waves are observed, which vary from 1 mm to 5 mm at the pile head. The horizontal deformation of the RPC pile increases gradually under sinusoidal wave loads from 1 Hz to 4 Hz, but decreases significantly under sinusoidal wave loads from 8 Hz to 16 Hz. The deformations of the pile are more significant when the RPC pile is subjected to a sinusoidal wave of

4 Hz and 8 Hz. However, the deformations of pile are more significant under the frequencies of 2 Hz and 4 Hz for the RC pile. It was because the natural frequencies of RPC and RC piles are inconsistent, that the first natural frequency of RPC ranges from 4 Hz to 8 Hz, and that of the RC pile is between 2 Hz and 4 Hz. In addition, the horizontal deformations of RPC and RC piles are 4.5 mm and 5.0 mm, respectively, at the pile head. Consequently, it concluded that the deformation of the RC pile is larger than that of the RPC pile, but the increment is slight.

### 3.4.2. The Influence of Seismic Waves

The horizontal deformations of micro piles under different seismic loads in cases 6, 7, and 8 (EI-Centro wave, Kobe wave, and artificial wave with amplitude of 0.15 g, respectively) were plotted in Figure 12. It can be clearly seen from Figure 12 that the horizontal deformation distribution laws of the pile along its buried depth are consistent basically for each pile. The maximum deformations of piles under different seismic wave loads are observed at the pile head. It indicated that the horizontal deformations of piles above the soil surface decrease linearly from the pile head to the soil surface. Moreover, the horizontal deformations of piles body decrease significantly with the increase of buried depth. Then, the horizontal deformations of piles reduce to 0 mm at the buried depth of 0.4 m (4D) and then change into the negative value. The maximum negative deformations of the piles are observed at the buried depth of 0.8 m (8D), and it gradually diminishes to 0 mm as the buried depth increases.

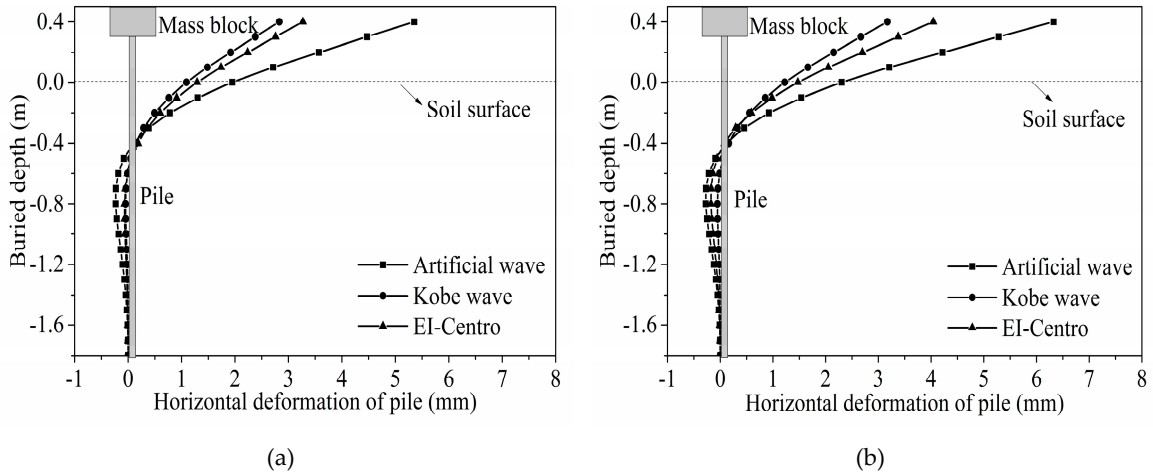

(a)    (b)

**Figure 12.** Horizontal deformations of piles under different seismic wave loads. (**a**) RPC micro pile. (**b**) RC micro pile.

As seen in Figure 12, it is also found that the horizontal deformation of piles under artificial wave load is greater than that of the Kobe wave or the EI-Centro wave. Especially, the maximum deformation of RPC and RC piles reach 5.5 mm and 6.3 mm at the pile head, but the deformations of piles are 3.1 mm and 3.2 mm, respectively, under the Kobe wave load. Therefore, the horizontal deformation of the RC pile is larger than that of the RPC pile, which is increased by 15% when piles are subjected to the artificial wave. Besides, comparing Figures 11 and 12, it indicates that the negative deformations of piles under seismic wave loads are more significant than that under the sinusoidal wave loads. The reason is that the frequencies of the sinusoidal wave were monotonous whereas the frequencies of seismic waves were abundant. The frequency of the seismic wave with abundant frequency includes the frequency component corresponding to the higher-order vibration modes of the micro pile. Although the horizontal displacement of the RC pile is larger than that of the RPC pile, but the increment is slight, at only 15%. It is noteworthy that the RPC micro pile has better crack resistance, higher ductility, and flexural rigidity than that of the RC pile.

## 4. Observations and Conclusions

To further discuss the dynamic response characteristics of micro pile-soil interaction, the shaking table tests on dynamic response of RPC and RC micro piles have been carried out in this paper. Meanwhile, the natural frequencies, the strains time history of micro piles-soil interaction, the bending moments and deformations of piles have been obtained under sine waves with different frequencies and seismic wave loads. The experimental results are described as follows.

(1) The first and second natural frequencies of the RPC micro pile are about 4.7 Hz and 16.3 Hz, and that of RC are 3.62 Hz and 15.4 Hz, respectively. Moreover, both compactness of sand and the natural frequencies of micro piles are increased under the load of repeated white noise.

(2) Compared with the action of sine waves and the action of seismic waves, the frequency of sine wave is relatively monotonous, but the frequency of the seismic wave is abundant, inducing the reverse deformation and bending moment of the pile more remarkable under the dynamic excitation load. It is also found that the dynamic response characteristics of piles under artificial wave were the most significant, which was followed by the EI-Centro wave, and then by the Kobe wave. The responses of piles under seismic wave loads are larger than those of sine waves.

(3) The maximum strains of piles were observed at the depth of 4.2 D (D is the diameter of the pile). Meanwhile, the maximum bending moments of the RPC and RC pile appear at the depth of 0.64D and 0.42 D, respectively, under a dynamic excitation. The peak horizontal deformations of piles were demonstrated at the pile head.

(4) Compared with the dynamic responses of the RC pile, it can be seen that the bending moments and the strain responses of RPC are larger than that of the RC pile, and the maximum increased by 40% and 98%, respectively. In addition, although the horizontal displacements of the RC pile are larger than that of the RPC pile, the increments are slight at only the range of 15%. Therefore, the RPC micro pile with steel fiber has better crack resistance, higher ductility, and flexural rigidity than that of the RC pile.

In addition, due to the relatively small amount of specimen, as well as the influence of artificial factors such as specimen fabrication, the measuring method, and so on. A large number of experimental data points need to verify the dynamic response performances of micro pile-soil interaction under shaking table tests.

**Author Contributions:** All authors substantially contributed to this work. J.C. and L.X. designed and fabricated the micro-pile model. J.C., Y.Z., and L.X. designed the experiment. J.C. and L.X. performed the experiments, analysis. J.C., X.L. (Xiaoyong Luo) wrote the paper and X.L. (Xiaoye Luo) revised and finalized the paper. All authors helped with the writing of the paper.

**Acknowledgments:** The authors gratefully acknowledge the financial supports provided by the Natural Science Foundation of China with Grants Nos. 51778147, the National key fundamental Research and development Funds for the 13th five-year Plan (2016YFC0701700), and the Hunan Provincial Innovation Foundation for Postgraduate.

**Conflicts of Interest:** The authors declare no conflict of interest.

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
