# Peer review of "Experimental Study on Dynamic Response Characteristics of RPC and RC Micro Piles in SAJBs"

_applsci, doi:10.3390/app9132644_

Round 1
Reviewer 1 Report
The paper is of interest for the journal audience.
The work is clearly presented and the results are interesting.
After a careful editing (language and typos) the paper can be accepted for publication.
Author Response
Reviewers1' comments:
Point 1: The paper is of interest for the journal audience.
The work is clearly presented and the results are interesting.
After a careful editing (language and typos) the paper can be accepted for publication.
Response 1:
Dear reviewers,
Thank you very much for your comments. Your suggestions are greatly helpful and beneficial to the improvement of the manuscript. A part of corrections (include language and typos) in the paper are listed in upload manuscript as following:
1. Line 19 (page 1), the statements of “Response Characteristics” were corrected as “response characteristics”, “good” was replaced as “better” in line 26.
2. Line 45 (page 2), “to improve the seismic behaviour of SAJB” was added, and “it was” was corrected as “They were”.
3. Line 47 (page 2), “structure”, “and subsidence” were added respectively, and “increase” was corrected “enhance”.
4. Line 62 (page 2), “The most completed book on micro-pile have been published by” and “in practical engineering” were added.
3. Line 70 (page 2), “Chen [19] and Farina [20] have done some works on the dynamic responses of structure-soil interaction based on the shaking table tests. It is found……further investigated” was added. In addition, “were discussed in detail” was added, “desired” was replaced as “desirable” (line 86).
4. Line 90 (page 3), “In addition, there are numerous researches based on FEM analysis to simulate the interaction effect between micro pile and soil. Based on the Winkler foundation beam model, the soil…… such as NCHRP [27], API-RP2A [28] and Reese [29].” was added, and “and numerical simulation investigations” was added in line 99.
5.Line 144 (page 4), “cross-section” was deleted, and “reinforcing bars” was added. Besides, “to the inner of steel box walls” was added in line 201 (page 6).
6.Line 325 (page 11), “as plotted in Figure 8b. It can be seen that the bending moments ordering of RPC and RC micro piles are different under same frequencies loads.” was added. Meanwhile, “However, the first natural frequency of RPC is 4.74Hz, so that the bending moment responses are more obvious under the frequency loads of 4Hz and 8Hz. In addition,” was added in line 329.
7. Line 394 (page 13), “reserved” was deleted, and “negative”, “continuous” were added respectively.
8. Line 435 (page 15), “strains and acceleration responses of” was deleted, and “sine waves with different frequencies and” was added in line 437.
We tried our best to improve the manuscript and made some changes in the upload manuscript in detail. And here we did not list the changes but marked using the “Track Changes” function in revised paper. We appreciate for reviewer’s warm work earnestly, and hope that the correction will meet with approval.
Thank you very much for your comments and suggestions.

Reviewer 2 Report
In this study reactive powder concrete and reinforce concrete micro piles were designed and realized. The shaking table test on dynamic response was used to investigate the dynamic Response Characteristics (strain time history of pile-soil system, bending moment and deformation of piles).
The topic is interesting and well written.
The bibliography should be improved. A more extensive analysis should be done. See for example:
Guoxing, C., Su, C., Xi, Z., Xiuli, D., Chengzhi, Q. I., & Zhihua, W. (2015). Shaking-table tests and numerical simulations on a subway structure in soft soil. Soil Dynamics and Earthquake Engineering, 76, 13-28.
Fabbrocino, F., Farina, I., & Modano, M. (2017). Loading noise effects on the system identification of composite structures by dynamic tests with vibrodyne. Composites Part B: Engineering, 115, 376-383.
Good paper but some parts need to be improved.
The paper is not suitable for publication in the present form, it requires minor revision.
Author Response
Reviewers2' comments:
In this study reactive powder concrete and reinforce concrete micro piles were designed and realized. The shaking table test on dynamic response was used to investigate the dynamic Response Characteristics (strain time history of pile-soil system, bending moment and deformation of piles).
Point 1: The topic is interesting and well written.
The bibliography should be improved. A more extensive analysis should be done. See for example:
Guoxing, C., Su, C., Xi, Z., Xiuli, D., Chengzhi, Q. I., & Zhihua, W. (2015). Shaking-table tests and numerical simulations on a subway structure in soft soil. Soil Dynamics and Earthquake Engineering, 76, 13-28.
Fabbrocino, F., Farina, I., & Modano, M. (2017). Loading noise effects on the system identification of composite structures by dynamic tests with vibrodyne. Composites Part B: Engineering, 115, 376-383.
Good paper but some parts need to be improved.
The paper is not suitable for publication in the present form, it requires minor revision.
Response 1:
Dear reviewer,
Thank you very much for your comments. Your suggestions are greatly helpful and beneficial to the improvement of the manuscript. The literature review of the paper had been further improved and a number of recent notable researches were added in the update manuscript. A total of 11 literatures were added in the paper, they were listed as following:
[1] Hoppe, E., Weakley, K., Thompson, P. Jointless bridge design at the Virginia Department of Transportation. Transportation Research Procedia, 14 (2016): 3943-3952. (Section 1 introduction, line 36)
[2] Zhao, Q., Lin, C., Zhao, Y., Huang, G. Mechanical Characteristics of a New Type of Jointless Bridge with an Arch Structure. In 2018 3rd International Conference on Smart City and Systems Engineering (ICSCSE), (2018), December, 300-307. (line 36)
[3] Erhan, S., Dicleli, M. Comparative assessment of the seismic performance of integral and conventional bridges with respect to the differences at the abutments. Bulletin of Earthquake Engineering, 13(2) (2015): 653-677. (line 42)
[4] Wood, J. H. Earthquake Design of Bridges with Integral Abutments. In 6th International Conference on Earthquake Geotechnical Engineering (2015, November). (line 42)
[5] Guoxing, C., Su, C., Xi, Z., Xiuli, D., Chengzhi, Q. I., & Zhihua, W. Shaking-table tests and numerical simulations on a subway structure in soft soil. Soil Dynamics and Earthquake Engineering, 76 (2015):13-28. (line 70, page 2)
[6] Fabbrocino, F., Farina, I., & Modano, M. Loading noise effects on the system identification of composite structures by dynamic tests with vibrodyne. Composites Part B: Engineering, 115 (2017):376-383” (line 70).
[7] Makris, N., Gazetas, G. Dynamic pile‐soil‐pile interaction. Part II: Lateral and seismic response. Earthquake engineering & structural dynamics, 21(2) (1992): 145-162. (line 94, page 3)
[8] Gazetas, G., & Dobry, R. Horizontal response of piles in layered soils. Journal of Geotechnical Engineering, 110(1) (1984): 20-40. (line 94)
[9] National Cooperative Highway Research Program. Static and dynamic lateral loading of pile groups. NCHRP Report461, Transportation Research Board-National Research Council, (2001).665-669. (line 98)
[10] American Petroleum Institute. API recommended practice for planning, designing and constructing fixed offshore platforms (1976). (line 98)
[11] Reese, Lymon C. Laterally loaded piles: Program documentation: Journal of the Geotechnical Engineering division, American society of civil engineers, 103 (GT4) (1977) :287-305. (line 98)
The bibliography has been improved in upload manuscript. Meanwhile, a more extensive analysis on the corresponding literatures were conducted and added. In addition, we have made appropriate changes to the manuscript for language and spelling. And here we did not list the changes but marked using the “Track Changes” function in revised paper in detail. Please have a check.
We appreciate for reviewer’s warm work earnestly, and hope that the correction will meet with approval.
Thank you again.
